# Occurrence Prediction of Riffle Beetles (Coleoptera: Elmidae) in a Tropical Andean Basin of Ecuador Using Species Distribution Models

**DOI:** 10.3390/biology12030473

**Published:** 2023-03-20

**Authors:** Gonzalo Sotomayor, Jorge Romero, Daniela Ballari, Raúl F. Vázquez, Iván Ramírez-Morales, Henrietta Hampel, Xavier Galarza, Bolívar Montesinos, Marie Anne Eurie Forio, Peter L. M. Goethals

**Affiliations:** 1Department of Animal Sciences and Aquatic Ecology, Faculty of Bioscience Engineering, Ghent University, Coupure Links 653, 9000 Ghent, Belgium; marie.forio@ugent.be (M.A.E.F.); peter.goethals@ugent.be (P.L.M.G.); 2Departamento de Ingeniería Civil, Facultad de Ingeniería, Universidad de Cuenca, Av. 12 de abril S/N, Cuenca, Azuay 010203, Ecuador; raulfvazquezz@yahoo.co.uk; 3Instituto de Estudios del Régimen Seccional del Ecuador (IERSE), Facultad de Ciencia y Tecnología, Universidad del Azuay, Cuenca 010204, Ecuador; jaromero@uazuay.edu.ec (J.R.); dballari@uazuay.edu.ec (D.B.); xavier.galarzag@hotmail.com (X.G.); 4Laboratorio de Ecología Acuática (LEA), Facultad de Ciencias Químicas, Universidad de Cuenca, Av. 12 de abril S/N, Cuenca 010203, Ecuador; hennihampel@gmail.com; 5DINTA Research Group, Universidad Técnica de Machala, Machala 070213, Ecuador; iramirez@utmachala.edu.ec; 6Ministerio del Ambiente, Agua y Transición Ecológica, Dirección Zonal 6, Cuenca 010104, Ecuador; bolivar.montesinos@ambiente.gob.ec

**Keywords:** Elmid genera, presence–absence records, species distribution models, random forest, streams

## Abstract

**Simple Summary:**

A machine learning algorithm, Random Forest, was used to establish species distribution models for five riffle beetle genera (Elmidae) in the Paute river basin (southern Ecuador), considering meteorology, land use, hydrology, and topography as environmental/explanatory variables. Alterations to riparian vegetation, canopy presence/absence, precipitation, elevation, and slope accounted for most of the Elmidae spatial variability. Clean and healthy streams were predicted to be the most likely places for Elmidae genera to occur. Additionally, specific ecological niches were predicted for each Elmidae genus. These findings can contribute significantly to conservation and restoration efforts in the study basin and could have implications for similar eco-hydrological systems.

**Abstract:**

Genera and species of Elmidae (riffle beetles) are sensitive to water pollution; however, in tropical freshwater ecosystems, their requirements regarding environmental factors need to be investigated. Species distribution models (SDMs) were established for five elmid genera in the Paute river basin (southern Ecuador) using the Random Forest (RF) algorithm considering environmental variables, i.e., meteorology, land use, hydrology, and topography. Each RF-based model was trained and optimised using cross-validation. Environmental variables that explained most of the Elmidae spatial variability were land use (i.e., riparian vegetation alteration and presence/absence of canopy), precipitation, and topography, mainly elevation and slope. The highest probability of occurrence for elmids genera was predicted in streams located within well-preserved zones. Moreover, specific ecological niches were spatially predicted for each genus. *Macrelmis* was predicted in the lower and forested areas, with high precipitation levels, towards the Amazon basin. *Austrelmis* was predicted to be in the upper parts of the basin, i.e., páramo ecosystems, with an excellent level of conservation of their riparian ecosystems. *Austrolimnius* and *Heterelmis* were also predicted in the upper parts of the basin but in more widespread elevation ranges, in the *Heterelmis* case, and even in some areas with a medium level of anthropisation. *Neoelmis* was predicted to be in the mid-region of the study basin in high altitudinal streams with a high degree of meandering. The main findings of this research are likely to contribute significantly to local conservation and restoration efforts being implemented in the study basin and could be extrapolated to similar eco-hydrological systems.

## 1. Introduction

Freshwater ecosystems have been severely altered by human activities and are significantly vulnerable to climate change [1,2]. Hence, there is an urgent need to understand the spatial and temporal patterns of aquatic organisms to maintain and restore aquatic biodiversity [3]. In this context, species distribution models (SDMs) relate taxa occurrence with the local environmental conditions and provide a spatial prediction of taxa habitat suitability on the entire study area and, optionally, across time [4]. Thus, SDMs can be particularly useful for biodiversity conservation policies, since they can identify suitable areas for preserving threatened taxa or priority areas for future sampling efforts [5]. SDMs have been used for identifying trends of spatial distribution and habitat limitations for some specific aquatic organisms, e.g., benthic macroinvertebrates [6,7,8], fishes [9,10], and algae [11]. Furthermore, SDMs could help manage water resources in a country such as Ecuador, which is facing a severe decline in the ecological integrity of its rivers and lakes [12,13]. However, to our knowledge, just one work has been carried out in the country using the SDMs framework, particularly targeting benthic macroinvertebrates taxa [14].

In this context, riffle beetles (i.e., Coleoptera: Elmidae) are cosmopolitan freshwater coleopterans that inhabit clean and well-oxygenated running waters in their larval and adult stages. Regarding ecosystem functionality, the Elmidae members are collector-gatherers and scrapers that feed mainly on algae and detritus. Further, elmid larvae and adults are an important part of the diet of fish. Elmidae taxa are sensitive to changes in the habitat structure and physicochemical conditions of aquatic ecosystems [15,16,17]. Therefore, elmids are considered excellent indicators of water quality integrity and perhaps also of climate change [18].

In the Paute river basin (PRB), which is one of the most important hydrological systems of Ecuador owing to its significant hydroelectric potential [19,20], elmids have been identified as the key taxa to establish adequate stream ecohydrological characterisation [21]. Notwithstanding the usefulness of elmids as bioindicators of freshwater ecosystems integrity in tropical zones [22], there is little research that focuses on the individual ecological requirements of the set of genera that the Elmidae family encapsulates. The problem with higher taxonomic resolution data is that they include several lower resolution taxa, which may have different environmental/ecological preferences. Thus, working on higher taxonomic resolution (e.g., Elmidae family) may mask the ecological sensitivities of taxa of lower resolution (e.g., genera of Elmidae) [23]. In this context, assessing the suitable habitats of different elmid genera is important to drive key study site conservation and restoration efforts. Thus, one way to cover the lack of knowledge about individual elmid ecohydrological preferences is through SDMs.

Further, while the species modelling framework is similar in the terrestrial, marine, and freshwater realms, each realm comprises specific challenges for combining the spatial scale, the environmental data, and the species records for building reliable models [24]. Thus, the choice of the modelling tool is an essential aspect of the development of SDMs. Worldwide, the Maximum Entropy Algorithm [25,26] is the most used tool for developing SDMs using the MaxEnt software [27]. Nevertheless, considering some common negative features of the SDMs, mainly dealing with the class imbalance nature of SDMs [28] and the availability of too few samples in large under-sampled areas [29], the use of the Random Forest (RF) algorithm is an attractive alternative [30,31,32]. Correspondingly, the current research echoes this latter trend by using the RF algorithm to model the occurrence probability of the elmid genera in the study basin. Within this frame of reference, the general goal of the current research was to develop and assess the SDMs models of riffle beetles (Coleoptera: Elmidae) in the Paute river basin. The main specific goals of the current research were (1) building different SDMs for five genera of Elmidae recorded in the study basin, (2) identifying the most important environmental factors that explain the spatial distribution of the elmid genera, and (3) performing a congruence assessment of the different SDMs of the study elmid genera.

## 2. Materials and Methods

### 2.1. Study Area 

The Paute river basin (PRB), in the south of Ecuador (Figure 1), has an area of 6442 km^2^, including the eastern lower portion towards the Amazon plateau. Its elevation ranges between 410 and 4687 m above sea level (a.s.l.), and slopes vary between 25% and 50%. The lower temperatures correspond to the western Andes range with a mean daily value of about 6 °C (at about 3500 m a.s.l.), while the warmest areas are situated in the Amazonian-influenced valleys and subtropical zones, with a mean daily value of 24 °C; nevertheless, a remarkable diurnal amplitude was observed. Due to the altitudinal gradient, mean annual rainfall oscillates in intensity and duration, with the lowest value of 660 mm at the basin’s centre and the highest observed value exceeding 3400 mm near the basin outlet. On the other hand, meteorological stations located at higher elevations (above 3000 m a.s.l.) receive between 1000 and 1400 mm [33]. Two major cities, namely Cuenca and Azogues, are in the basin, with approximately 600,000 and 40,000 inhabitants, respectively. Important conservation zones are in the study basin, the most relevant (Figure 1) the Cajas National Park (CNP) and the Sangay National Park (SNP), both UNESCO World Heritage Sites. However, despite these conservation efforts, domestic wastewaters, agricultural runoff, animal husbandry, and industrial effluents are negative factors that are known to influence the surface water quality (WQ) of the study basin [21,34,35].

### 2.2. Sampling of Riffle Beetles

The benthic macroinvertebrate community was sampled at 67 sites located in the study basin throughout four years (2010–2012 and 2015) by the former Ecuadorian National Secretary of Water (SENAGUA), Santiago River Hydrographic Demarcation (DHS), and the Municipal Public Enterprise of Telecommunications, Drinking Water, Sewerage and Sanitation of Cuenca (ETAPA EP). Samples were collected using a D-frame kick net (25 cm aperture, 0.5 mm mesh) [36]. Sampling encompassed all existing microhabitats characterised by different depths, substrates, and water velocities. Macroinvertebrate samples were preserved in 70% ethanol and sorted using a stereomicroscope. Using these samples, the presence–absence data records of elmids were obtained (Figure 2). The sampling sites were visited four times per year (on average). Some were sampled more frequently because they were located either at highly impacted sites or, on the contrary, at unaltered environmental (i.e., reference) locations.

#### Riffle Beetles and Their Presence–Absence Records

A total of 1672 elmid records were compiled and grouped into five genera (i.e., n_gen_ = 5) that belong to the subfamily Elminae [37], namely, *Austrelmis* (g_1_, 8.4%), *Austrolimnius* (g_2_, 26.6%), *Heterelmis* (g_3_, 30.3%), *Macrelmis* (g_4_, 20.0%), and *Neoelmis* (g_5_, 14.7%). The research was limited to genera as most records of Elmidae from the study basin were predominantly larvae, which can only be identified at the genus level [38]. Herein, to use a record of an elmid genus, such as presence data, to perform the modelling process, the minimal sample size [39] was greater than two individuals per taxa. The latter was carried out to minimise the probability that an individual of a given taxa was recorded accidentally (i.e., fortuitous arrival through a strong current or a dead individual drifted downwards by the river current, etc.) in the sampling station of interest.

### 2.3. Environmental Variables

This study used twelve environmental variables (12en_v_) as the (independent) descriptive factors to explain the spatial variability of elmid genera. The twelve variables (Table 1) were selected from a previous set of 20 variables (en_v_) upon a Pearson’s correlation analysis that enabled excluding redundant en_v_ characterised by positive or negative correlation magnitudes above 0.75 [40]. This was done to achieve a parsimonious model and to minimise the risk of overfitting it. The correlation analysis was performed with the R package ‘ENMTools 1.0’ [41]. The eight excluded en_v_ were: solar radiation, roughness index, stream power index, flow accumulation, Strahler stream order, canopy height, evapotranspiration, and environmental temperature.

The unit of analysis for this research was the hydrographic network of the basin generated from a Digital Elevation Model (DEM), a LIDAR product of the SIGTIERRAS project (http://www.sigtierras.gob.ec accessed on 7 February 2022) of the Ecuadorian government [42]. Its original horizontal resolution is 3 m, whilst its vertical precision is ±1.5 m. However, to reduce computational running times to reasonable levels, its horizontal resolution was resampled up to 12 m using the Bilinear algorithm available in the Resampling set of tools of ArcGIS 10.4.1 software [43,44]. The respective resampled product (DEM_r_) was used in the rest of the analysis.

The hydrographic network (Hy_net_) was obtained using the Hydrology toolbox of ArcGIS 10.4.1, which applies the method for extracting hydrographic networks from DEMs [45,46]. Thus, the following steps were applied: (1) pre-processing the DEM_r_; (2) determining the flow direction; (3) calculating the cumulative amount of the flow confluence; (4) determining the confluence threshold; and (5) generating the hydrographic network [47,48]. Hereafter, the spatial distribution of each one of the 12en_v_ was incorporated into a raster layer that was previously cropped according to the Hy_net_ mask, producing a continuum of environmental predictors along the stream network [49].

Eastness (East) and northness (Ntns) provide continuous measures describing geographical orientation in combination with slope. For the Northness, +1 represents the north and −1 south directions. For Eastness, +1 represents the east and −1 the west directions [68]. Sinuosity (Snty) provides the degree of meandering of the stream channel. In general, Snty = 1 is linked to a straight channel, and Snty = 4.8 is the maximum degree of meandering in the Paute river basin hydrographic network. The Lithology (Ltlgy) variable implies 78 lithological groups for the Paute river basin. The first half of these 78 groups correspond to sands, sandstones, clasts, and schists; the second half corresponds to silts, clays, pyroclasts, and undifferentiated metamorphic rocks. The soil type (Soils) variable accounts for the ten soil units that exist in the study basin, i.e., Andisols (1), Inceptisols (2), Mollisols (3), Vertisols (4), Entisols (5), Alfisols (6), Oxisols (7), Histosols (8), Ultisols (9), and miscellaneous (10). Canopy (Cnpy) ranges from 1 to 100, with 1 representing riparian areas without the presence of forest and 100 riparian areas with high forest presence.

### 2.4. Species Distribution Models (SDMs) Using Random Forest (RF) Algorithm

The Random Forest (RF) algorithm [69] is an ensemble of classification or regression trees and is widely used in research, including SDMs analyses [70]. It performs classification analysis by building many decision trees from bootstrap data set samples. The final model prediction is performed by averaging the predictions made by each tree in the forest. In this study, RF was implemented using the R Package ‘Biomod2’ [71] to model the spatial distribution of the presence–absence of elmid genera (i.e., occurrence probability) as a function of the 12en_v_.

Each elmid genus was separately modelled, i.e., five RF modelling processes were performed. The tuned parameters to estimate the different RF models were the number of trees (n_tree_) and the number of variables randomly selected at each node (mtry), given that the RF algorithm is prone to be sensitive to these parameters [72,73]. Herein, for parameterising the RF algorithm, the strategy of Strobl et al. [74,75] was implemented. It was based in a grid search through which all possible combinations of given discrete parameter regions were evaluated. Values of mtry = 5 and n_tree_ = 3000 were adopted in this study after a sensitivity analysis that showed more consistent results with these values.

The different RF runs were carried out using the K-fold cross-validation (CV) method, in which the data were divided into K disjoint sets (folds), and the K-th fold was used as an independent test (i.e., validation) set. The remaining (K –1) folds were used to train the RF model and find its different parameters, after which model validation took place using the test set. This process was repeated n times. The error estimation was averaged over all n trials to get the total effectiveness of the model [76]. For the current research, K = 4 with n = 3 repetitions was used for each elmid genera, producing twelve runs (models) for each of the 5 genera. Trade-offs were involved when selecting K number of folds [77]. Using K = 4 (implying the use of 75% of the data for training and 25% for validation) has been reported as an excellent value to perform a realistic classification assessment [78].

Since the available response variables, i.e., presence–absence records of elmid genera, were imbalanced (Figure 2), RF was chosen for this study because it is known to work well with imbalanced data sets in a classification framework [79,80,81]. For further details about RF, the reader is kindly referred, for instance, to [82,83].

For evaluating the RF outputs, the area under the receiver operating characteristic (ROC) curve was used, which was applied for the analysis of classification performances in the framework of binary classification of samples as positive (P) or negative (N). In this context, the ROC curve is defined [84] as a plot of x = 1 − Sp_P_ (specificity of the positive class, also known as False Positive Rate, FPR) versus y = Sn_P_ (sensitivity of the positive class, also known as True Positive Rate, TPR). Given the ROC curve for a classifier, the area under the curve (AUC) measures its overall diagnostic performance, with AUC = (Sp_P_ + Sn_P_)/2 [85]. Since the AUC is a portion of the area of a unit square, its value varies between 0 and 1, with 1 being its optimal value. For each elmid genus, there were 12 output models because of the cross-validation process; thereby, 60 AUC values (i.e., n_gen_ x 12) were obtained in total. For each elmid genus, its 12 AUC values were aggregated into a single value using central tendency measures. Before this aggregating process, the normality of each set of 12 AUC values was checked [35,86] using the Shapiro–Wilk (S–W) test [87] considering a 95% confidence level. For a particular elmid genus, if the S–W test suggested normality, the mean AUC value was used for aggregating; otherwise, the median was assigned as the aggregated AUC value [88,89]. For the interpretation of the AUC values, it followed the proposal of Hosmer et al. [90], where an AUC = 0.5 could be interpreted as “no discrimination”; 0.5 < AUC < 0.7 as “poor discrimination”; 0.7 ≤ AUC < 0.8 as “acceptable discrimination”; 0.8 ≤ AUC < 0.9 as “excellent discrimination”; and AUC ≥ 0.9 as “outstanding discrimination”.

#### Assessing Significant Environmental Variables

The ‘Biomod2’ R package [91] uses a random permutation procedure to estimate the importance (var_imp_) of each 12en_v_. The procedure (Figure 3) is independent of the modelling technique. It uses Pearson’s correlation between the standard prediction (i.e., fitted values) and the predictions obtained by focusing the simulations on a given environmental variable and randomly permutating its value for every simulation. If the correlation is high, i.e., showing little difference between the standard and a given prediction, the given variable is considered unimportant for the model. This is repeated several times for each given variable, and Pearson’s mean correlation coefficient over the runs is kept. Herein, the number of permutations to estimate the var_imp_ for every one of the 12en_v_ was 5.

As a result, the R ‘Biomod2’ package produces a ranking of variables and their corresponding var_imp_ values. In this context, for each elmid genus, there were 12 output models because of the cross-validation process, thus 144 var_imp_ values (i.e., 12en_v_ x 12 output models). In five separate analyses (i.e., one for each genus), for each 12en_v_, their 12 var_imp_ values were aggregated. Thus, as in the case of AUC, before the aggregating process took place, the normality trend of each set of 12 var_imp_ values was checked using the S–W test, considering a 95% confidence level. For each elmid genus, their aggregated var_imp_ values from each 12en_v_ were expressed as percentages and ranked in descending order. However, there is not a statistically based var_imp_ threshold on distinguishing between important and non-important en_v_ to explain the spatial distribution of elmids. Thus, a variable segregation analysis was carried out for each genus. In this analysis, the important set of en_v_ (en_v-imp_) was identified by removing, one by one, the non-important en_v_ with respect to the (standard) RF-based model containing all the 12en_v_. On every occasion, after a given en_v_ was removed from a previous RF-based model, the complete modelling approach was repeated so that a cross-validation analysis was entirely performed, and the respective aggregated AUC value was obtained for the newer model. This variable segregation approach was carried out until the best RF-based model, formed by the set of en_v-imp_, was identified by the highest aggregated AUC value. In this analysis, variables were removed by considering the ranked var_imp_ information so that the en_v_ associated with lower var_imp_ was removed first.

Additionally, for each en_v-imp_ of elmid genera, response curves were created using the AUC data. Thus, to define the optimal range for the distribution of each genus per each en_v-imp_, a cluster analysis through the k-means method was implemented using the AUC values. With this procedure, it is possible to distinguish the statistical cut-off AUC values (i.e., borders or thresholds) and, thereby, the optimal range of preference of each genus for each en_v-imp_. K-means clustering is a non-hierarchical clustering method that assigns each object to the group with the closest centroid by calculating the centroid of each group [92]. This study applied this method using the Euclidean distance as the similarity measure between objects. In the k-means algorithm, the number of clusters is specified a priori, usually according to some hypothesis [93]; however, a more robust statistical procedure uses internal validation indices [94]. Using quantities and features inherent in the data, an internal index measures the appropriateness of clustering partitions without external information [95]. Herein, an internal validity index was applied, namely, the Silhouette Coefficient (SC). The Cluster Validity Analysis Platform (CVAP) was used for this purpose [94]. SC [96] is a dimensionless measure that evaluates the quality of compactness and separation of clusters; with an upper bound equal to 1, the optimum k value corresponds to its largest average. The inspected number of clusters k was from 2 to 5.

### 2.5. Prediction of Spatial Distribution

In the ‘Biomod2’ R package, the final generated models using the environmental space (en_v-imp_) were projected within the Hy_net_ to create the spatial predictions for each elmid genera [97]. These SDMs contain the occurrence probability values for each elmid genera. Correspondingly, values close to 0 indicate probable absences, and values close to 1 suggest probable presences. The twelve SDMs outputs for each genus of the Elmidae family were exported in ESRI raster format (GRID) to facilitate their processing/averaging using the Raster Calculator tool available in ArcGIS 10.4.1 [91]. As a result, one final SDM for each elmid genus was created, i.e., SDM_g1_, …, SDM_g5_. Further, to improve the visualisation of these SDMs, each one of them was reclassified considering three probability classes of the spatial occurrence of modelled taxa, i.e., low (C1, 0–0.33), medium (C2, 0.3–0.66) and high (C3, 0.66–1). This number of classes was chosen following a previous study (Sotomayor et al., 2020, 2021) that concluded that three classes are adequate for characterising the water quality in the Paute river basin.

### 2.6. Congruency of the Predicted Spatial Distribution of the C3 Probability of Occurrence of Elmid Genera

The congruency of the C3 class of occurrence probability of elmid genera was assessed by its visual comparison with the land use (LU)–land coverage (LC) distribution in the study basin [98]. The LU–LC data were reclassified to describe the spatial distribution of the anthropogenic impact (higher or lower) level, which was compared to the distribution of the C3 class of occurrence probability of elmid genera. This latter distribution was obtained by merging the respective spatial distribution of every one of the five study genera. The original LU–LC classes [98] were the following: (1) altered vegetation; (2) woody native vegetation; (3) without cover/urbanised, (4) páramo ecosystem; and (5) water. The higher anthropogenic impact class was defined upon the reclassification of LU–LC classes 1 and 3, whilst the lower anthropogenic impact class was defined upon classes 2, 4, and 5. Thus, these anthropogenic impact classes are not the result of any additional calculation of an index or a factor but just a simple reclassification of the original LU–LC information. ArcGIS 10.4.1 was used for all the respective Geographic Information Systems (GIS) analyses.

## 3. Results

### 3.1. Species Distribution Models (SDMs)

A significantly outperforming RF (i.e., aggregated AUC values) was observed when only the informative environmental variables (i.e., en_v-imp_) were used in the modelling process. The statistical performance of the RF models, i.e., the aggregated AUC values linked to each SDM of elmid genera (Table 2), suggested that the RF model for *Austrolimnius*, *Austrelmis*, and *Macrelmis* had the best performance (i.e., excellent discrimination), followed by *Heterelmis* and *Neoelmis* (i.e., acceptable discrimination) [90]. The aggregated AUC values ranged from 0.76 to 0.89 (Table 2). The spatial extent for each genus of the Elmidae family upon their probability of occurrence ranges (i.e., C1, C2, and C3) indicated that *Austrolimnius* and *Heterelmis* are the taxa with the most widespread spatial probability of occurrence in the study site. *Macrelmis* would occur, in its great majority, in the lowest basin areas, toward the Amazon basin. On the contrary, *Austrelmis* is likely to occur in the higher elevations of the basin, especially in the protected zones such as the Cajas National Park (CNP) and the Sangay National Park (SNP). *Neoelmis* shows low and medium probabilities of occurrence in the studied basin (Table 2, Figure 4).

### 3.2. Assessing Significant Environmental Variables

The en_v_ that showed the highest association with the spatial distribution of Elmidae were slope, eastness, elevation, precipitation, Shreve stream order, lithology, canopy, percentage of riparian alteration, flow direction, and sinuosity (Table 3). Northness and soil types were non-important variables. The response curves of the important variables (en_v-imp_) for each genus and their optimal probability range of preference are presented in Figure 5. The first en_v-imp_ for all genera of the Elmidae family was the most important for modelling the spatial distribution of the occurrence probability of a given elmid genus (Table 3). The curves of the first en_v-imp_ differed from the symmetric bell-shaped form. They exhibited clear peaks and depressions (Figure 5), indicating the variable ranges associated with higher (and lower) values of the probability of occurrence of elmid genera. Further, the en_v-imp_ that were lower in relevance (Table 3) had less discriminatory power in modelling the spatial distribution of the occurrence probability of a given elmid genus and, as such, exhibited fewer clear peaks and depressions (Figure 5).

Upon the AUC curves shown in Figure 5, the following environmental requirements for the Elmidae genera, reflected by the respective optimal probability of occurrence ranges, are distinguished. Higher values of the probability of occurrence for *Austrelmis* (g_1_) are in streams characterised by the environmental variable ranges: elevation [3111–3833] m a.s.l., precipitation [1279–1883] mm, eastness [0.49–0.99], slope [29–33]%, and riparian alteration [0–61]%. Higher probability values for *Austrolimnius* (g_2_) are in streams characterised by the ranges: elevation [3043–3833] m a.s.l., eastness [−2.7–0.990], flow direction [97.2–126.7], and slope [22.5–32.9]%. Higher probability values for *Heterelmis* (g_3_) are in streams characterised by the ranges: elevation [2734–3798] m a.s.l., lithology [43.3–75.3], slope [0.33–2.68]%, eastness [0.65–0.89], Shreve order [47.9–282.2], and riparian alteration [28.0–63.0]%. Higher probability values for *Macrelmis* (g_4_) are in streams characterised by the environmental ranges: precipitation [1093.4–2883.5] mm, Shreve order [1.0–422.9], elevation [433.0–3249.2] m a.s.l., eastness [−0.46–0.97], slope [23.8–32.9]%, and canopy [70.5–96.0]. Higher probability values for *Neoelmis* (g_5_), are in streams denoted by the ranges: precipitation [1279.4–2069.8] mm, slope [17.8–32.9]%, canopy [70.6–96.0], eastness [0.45–0.97], and sinuosity [1.1–1.4].

### 3.3. C3 Class of Occurrence Probability of Elmidae across the Paute River Basin

The spatial distribution of the C3 class of occurrence probability of Elmidae across the study basin and the respective distribution of the anthropogenic impact are shown in Figure 6. A lower anthropogenic level characterises 59.4% of the basin area, whilst the remaining 40.6% exhibits a higher anthropogenic level. The figure depicts that the C3 class of occurrence probability of Elmidae is, in average terms, not distributed in higher anthropogenic impacted zones, which is congruent with elmids being prone to be absent in (water quality) impacted zones.

## 4. Discussion

### 4.1. Model Selection

The use of the RF algorithm was successful in terms of the achieved modelling performance. Notwithstanding, most studies that chose the SDMs framework of analysis utilised the Maximum Entropy Algorithm [25,26] using the MaxEnt software [27] as their primary modelling tool since it has a user-friendly graphical interface (i.e., it is easy to use and enables a lucid visualisation of results) [99]. This trend of using MaxEnt has also been observed in Ecuador for spatially assessing organisms belonging to different biological communities [100,101,102,103,104,105,106]. However, some limitations of the MaxEnt modelling tool have been reported [107]. As it is the most relevant that this approach considers only presence (i.e., occurrence) data, this implies that the prevalence of the species (i.e., the proportion of occupied sites) cannot be precisely determined [108,109]. A second fundamental limitation of MaxEnt is that sample selection bias (whereby some areas in the landscape are sampled more intensively than others) has a much stronger effect on presence-only models than on presence–absence models. In this context, if the presence–absence survey data are available as in the current research, it is generally prudent to use a presence–absence modelling method [109], such as the RF algorithm, which has been tested as one of the most accurate tools for the construction of SDMs [29,30,31,32,110,111,112].

### 4.2. Model Performance

Some differences were observed in the performance of the RF algorithm (characterised by the AUC values) modelling the distribution of the five elmid genera (Table 2). Species distribution modelling quality could be assessed considering subjective AUC conditions/ranges. Alternatively, it could be adopted the AUC conditions/ranges suggested by Hosmer et al. [90] or the AUC condition (i.e., >0.7) used by Cha et al. [32] to regard the modelling quality as “excellent”. According to the first AUC criteria, the RF SDMs of *Austrelmis*, *Austrolimnius*, and *Macrelmis* could perform an overall “excellent discrimination”; the respective RF models of *Heterelmis* and *Neoelmis* performed an overall “acceptable discrimination”. In consonance with the second AUC criterion, given that the AUC values of all the RF SDMs of the study genera were higher or equal to 0.7, the overall performance of all the SDMs could be regarded as “excellent”. The False Positive Rate (FPR) confirms the AUC findings, in the sense that the RF-based SDMs for *Austrelmis* (19%), *Austrolimnius* (16%) and *Macrelmis* (19%) performed better than the respective models for *Heterelmis* (27%) and *Neoelmis* (34%).

Model performance is likely to be compromised using genera instead of species taxonomic level since genera can contain several different congeners, which could have different ecological requirements. However, all study genera prefer clean water conditions [34,35,86]; as such, it is assumed that congeners within each genus also have similar preferences. Hence, it is assumed that, under the current conditions, the model performance is only being compromised in a marginal level.

### 4.3. Basic Findings of the Developed SDMs

Elmidae family members are indicators of good stream ecosystem status [15,18,113,114]. The occurrence probabilities of elmids predicted in this study are notably lower in the human-impacted central region of the study catchment (Figure 4 and Figure 6). Further, some substantial differences in ecological requirements were predicted by the SDMs of elmid genera in the current research (Figure 4). These differences are like previous studies [16,22,115,116], which found that genera in the Elmidae family differ regarding their ecological requirements. In this context, despite some relevant works that have been published on the taxonomy of the Ecuadorian Elmidae in the last few years [117,118,119], no work has been done about species distribution modelling of Elmidae. Further, to the best of our knowledge, just one SDM study using MaxEnt considered Elmidae in a southern Brazilian basin [60]. Contrasting the findings of this study (Figure 4), Braun et al. [60] found very similar spatial distributions of the occurrence probability of their study genera. These differences in the outcomes of both studies are likely to be the consequence of (i) the different characteristics (elevation range, climate, geology, etc.) of the study basins and (ii) the different modelling approaches that were used in either study (i.e., presence–absence or presence-only modelling).

### 4.4. Important Predictors for Elmids Distribution

Despite the relatively similar identification of important variables for the models of the five study genera, each genus model has its own set of informative variables (Table 3, Figure 5). This finding emphasises the (predicted) dissimilarities of ecological requirements of some of the study genera.

González-Córdoba et al. [22] found in the Colombian Andean region that *Austrelmis* survive in a narrow temperature range of cold water. In contrast, *Austrolimnius* and *Heterelmis* tolerate a wide temperature range and survive cold and relatively warm water. Given the tight inverse relationship between elevation and temperature [120], both findings of González-Córdoba et al. [22] fit with the current research, where the higher probability of spatial occurrence of *Austrelmis* is between 3111 and 3833 m a.s.l. (Figure 4 and Figure 5), whilst *Austrolimnius* and *Heterelmis* tend to occur at higher and lower elevation streams (ranges between 3043 and 3833 and 2734 and 3798 m a.s.l., respectively). This highlights the importance of elevation for the spatial distribution of the *Austrelmis*, *Austrolimnius* and *Heterelmis*. The elevation preference for *Austrelmis* is like what has been estimated for the Cañete River basin in the south of Perú [121].

For *Austrelmis*, *Austrolimnius*, and *Heterelmis*, the lithology and the percentage of riparian alteration were selected as informative (Figure 5). Consistent with this finding, in the Ocoa river basin, Colombia, the conservancy of the riparian ecosystems has been reported as a critical aspect for *Austrolimnius* and *Heterelmis* [122]. It is likely to be the consequence of the fact that elmid members frequently use riparian forests for their terrestrial pupation [123]. Further, *Austrolimnius* and *Heterelmis* apparently prefer rivers located in formations such as silts, clays, pyroclasts, and undifferentiated metamorphic rocks; however, in the literature, there are no specific studies about the influence of this variable on Elmidae members. Overall, it is known that the macroinvertebrate communities are generally modified by local factors such as geology [124]; for example, the influence of the local geology was responsible for the high concentrations of salts in the Lincha river sub-basin (Perú), a factor that conditioned the presence or relative abundance of some taxa, including elmids [121].

The precipitation was another essential variable for the ecological requirements of *Macrelmis* and *Neoelmis* (Figure 5). In the case of *Macrelmis*, it is mainly predicted in the east of the study basin, where higher precipitations, in the range from 1090 to 2885 mm, are observed [125]. Because precipitation influences stream discharge, the latter is consistent with previous findings, as this genus is positively correlated with higher discharges [126].

Some studies found that *Macrelmis* is related to higher water temperatures [127,128,129]. Similarly, the respective SDM predicted a high probability of occurrence of *Macrelmis* towards the Amazon region where the temperature is higher. For *Macrelmis*, the canopy variable was important in the species distribution modelling process (Figure 5). Previous findings for *Macrelmis* suggest that they are closely related to areas with forested biomes [130], which is congruent with the findings of the current research as the high probability values for *Macrelmis* were predicted to occur in its great majority in the east of the basin (Figure 4), where high canopy values characterise sub-basins.

In the case of *Neoelmis*, the canopy variable was significant in its species distribution modelling (Figure 5), as members of this taxon were rarely predicted in the middle stripe of the study basin where forested areas are limited, and anthropogenic activities are important. González-Córdoba et al. [22] found that *Neoelmis* was present in a wide range of temperatures and even tolerates medium to high degrees of contamination. It is likely these differences are based on the hydrological systems and on the frameworks of research (i.e., the work of González-Córdoba et al. [22] is far from the concept of SDMs). Additionally, the possibility that different congeners vary in habitat preferences could explain the dissimilarity of the findings of *Neoelmis* between both studies. Just for *Neoelmis,* the sinuosity (i.e., Snty) was selected as an informative variable to explain its spatial distribution in the study basin, i.e., members of *Neoelmis* prefer streams with a high degree of meandering (Figure 5). However, although the importance of Snty to elmids has been described [131], for the specific case of *Neoelmis* no similar findings exist like the current research.

### 4.5. Elmidae Genera’s SDMs and Their Implications for the Surface Water Quality Management in the Study Basin

The high probability values of *Austrolimnius*, *Heterelmis*, and mainly *Austrelmis* are rarely predicted in most parts of the Burgay and Magdalena sub-basins (Figure 4), which have been described as polluted systems where the domestic and industrial wastewater discharge, extensive agriculture, cattle ranching, and the loss of native vegetation cover are the anthropogenic threats that cause severe surface water quality pollution and the subsequent loss of benthic macroinvertebrates taxa [21,34,35,132,133]. For the study basin, Sotomayor et al. [21,86] found that Elmidae is a keystone family in establishing adequate stream water quality assessments. Thus, despite the dissimilarity of genera in the Elmidae family regarding their ecological requirements, the overall trend of elmid members is that their occurrence probability in the study basin is higher in areas with good levels of conservation, e.g., protected areas (Figure 6). This is emphasised through the cross-check analysis of Figure 6, i.e., the high occurrence probability of Elmidae is less distributed in the areas with high anthropogenic impact. Likewise, in the high zones of the study basin, the anthropogenic activities are less than in their lower zones. That is, the biogeographical importance of the elevation for the potential distribution of elmids members is notorious. However, indirectly, anthropisation is a factor linked with the elevation, and both signals were detected in the current research.

## 5. Conclusions

Using the Random Forest algorithm, the genera of the Elmidae family were predicted in a great majority with good statistical reliability in healthy streams of the Paute river basin located in areas with good conservation status, i.e., protected areas. Some clear differences in ecological and environmental requirements were registered for some of the modelled elmid genera in the basin. The high probability values of spatial occurrence for *Austrelmis* are linked chiefly to streams of high mountains, i.e., the páramo ecosystems. Additionally predicted in the upper parts of the basin were *Austrolimnius* and *Heterelmis*, but at more varied elevation ranges, and *Heterelmis* in some areas where human activity was moderate. Contrary, *Macrelmis* were the majority predicted in the east of the studied basin in forest areas with high canopy values towards the Amazon system (also with high precipitation levels). According to predictions, *Neoelmis* would inhabit high altitudinal streams with a high degree of meandering in the middle portion of the Paute river basin. There is relative consistency in the informative variables that explain the spatial distribution of elmid genera in the study basin. Thus, factors such as elevation, precipitation, the quantity of water, and land use are linked to the general ecological requirements of the elmid genera and thereby to their occurrence probability. However, for each modelled genus, a specific set of environmental factors was observed, which implies dissimilarities in the ecological and biogeographical factors that govern the spatial distribution of the studied genera. However, despite these dissimilarities between elmid genera, the overall finding is that Elmidae members are good indicators of healthy freshwater ecosystems. The study revealed that the use of robust machine learning methods such as the one applied here, in conjunction with appropriate spatial analysis and visualisation tools, could be a promising approach to derive plausible geographical distributions of species (and genera) in support of conservation and management purposes, and that could be applied in other locations where suitable spatially distributed data are available.

## Figures and Tables

**Figure 1 biology-12-00473-f001:**
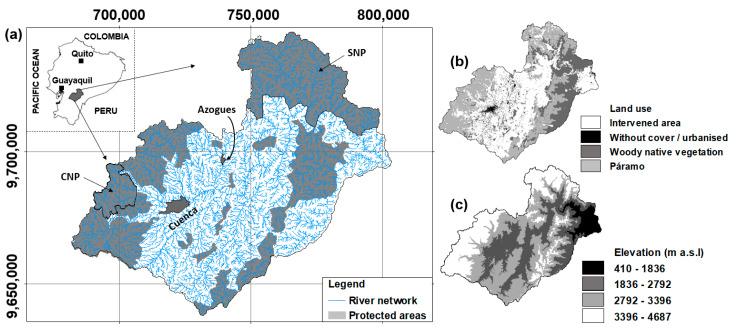
(**a**) Location of the Paute river basin in continental Ecuador and its two largest cities (Cuenca and Azogues); (**b**) main land cover classes distribution; and (**c**) elevation distribution in the basin. CNP = Cajas National Park; SNP = Sangay National Park. Coordinates system: WGS84 UTM 17S (m).

**Figure 2 biology-12-00473-f002:**
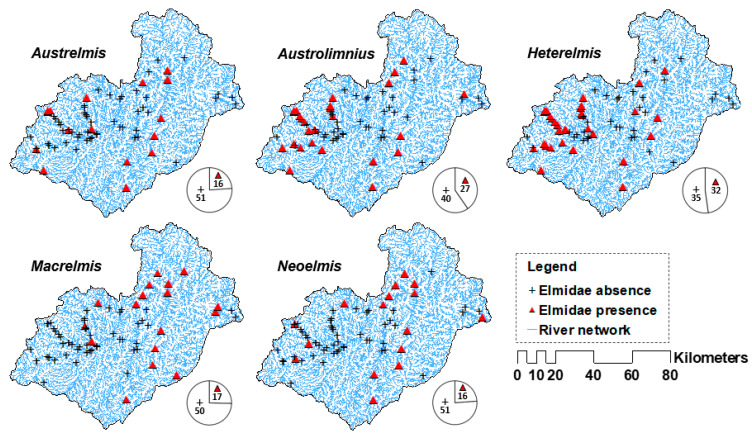
Spatial distribution of the observed presence–absence of the five genera of Elmidae records in the Paute river basin. The pie charts indicate the number of sampling sites with presence–absence records (the total number of sampling sites is 67).

**Figure 3 biology-12-00473-f003:**
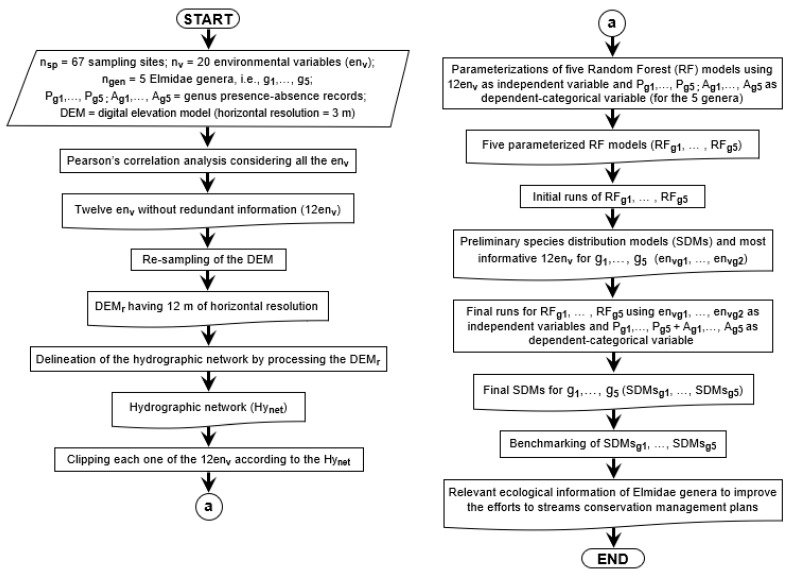
Flowchart of the methodology implemented in the current study.

**Figure 4 biology-12-00473-f004:**
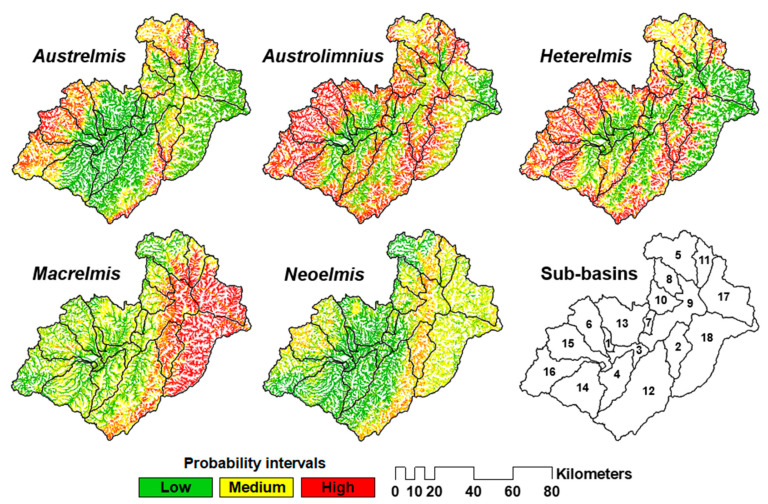
Spatial distribution of the probability of occurrence for the five genera of Elmidae in the Paute river basin, considering three classes, i.e., low (C1), medium (C2), and high (C3). Sub-basins: 1 = Sidcay, 2 = Collay, 3 = Cuenca, 4 = Jadán, 5 = Juval, 6 = Machángara, 7 = Magdalena, 8 = Mazar, 9 = Paute, 10 = Pindilig, 11 = Púlpito, 12 = Santa Bárbara, 13 = Burgay, 14 = Tarqui, 15 = Tomebamba, 16 = Yanuncay, 17 = Paute bajo, and 18 = Negro.

**Figure 5 biology-12-00473-f005:**
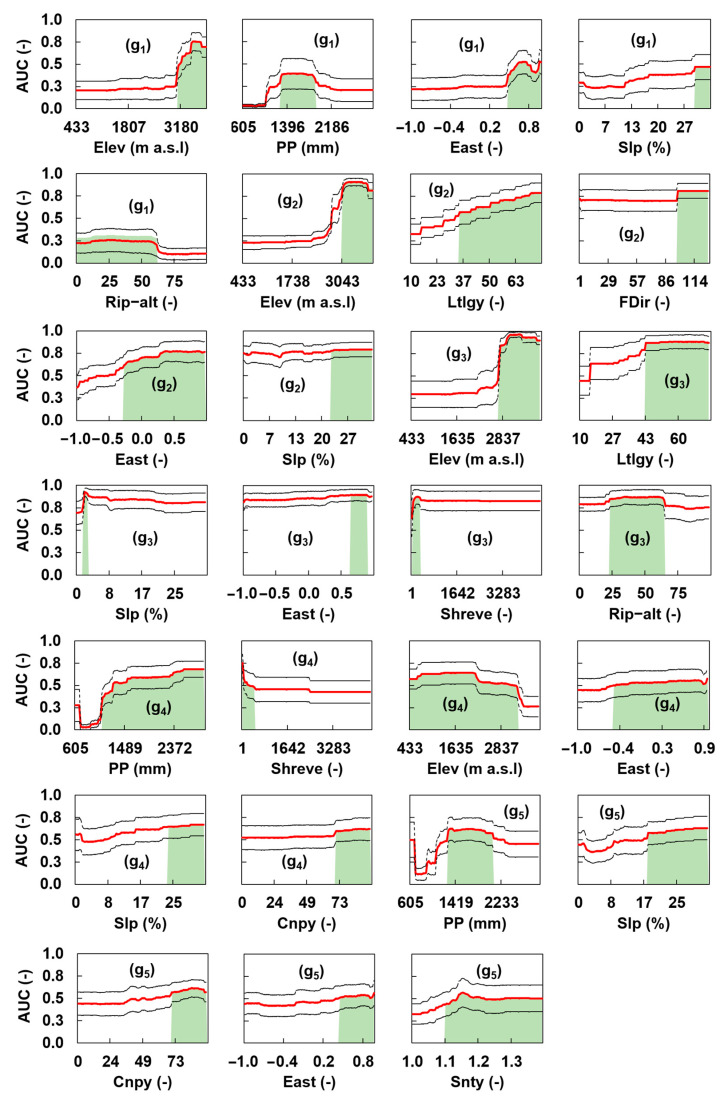
AUC response curves (red line) of each genus of the Elmidae family, namely: *Austrelmis* (g_1_), *Austrolimnius* (g_2_), *Heterelmis* (g_3_), *Macrelmis* (g_4_), and *Neoelmis* (g_5_) as a function of the important environmental variables (en_v-imp_). Dashed lines define the AUC standard deviation band. The highlighted area under the AUC indicates the optimal range of preference for the different Elmidae genera. Cnpy = Canopy, Elev = Elevation; East = Eastness, Fdir = Flow direction; Ltlgy = Lithology, PP = Precipitation, Rip-alt = Riparian alteration, Shreve = Shreve stream order, Slp = Slope, Snty = Sinuosity.

**Figure 6 biology-12-00473-f006:**
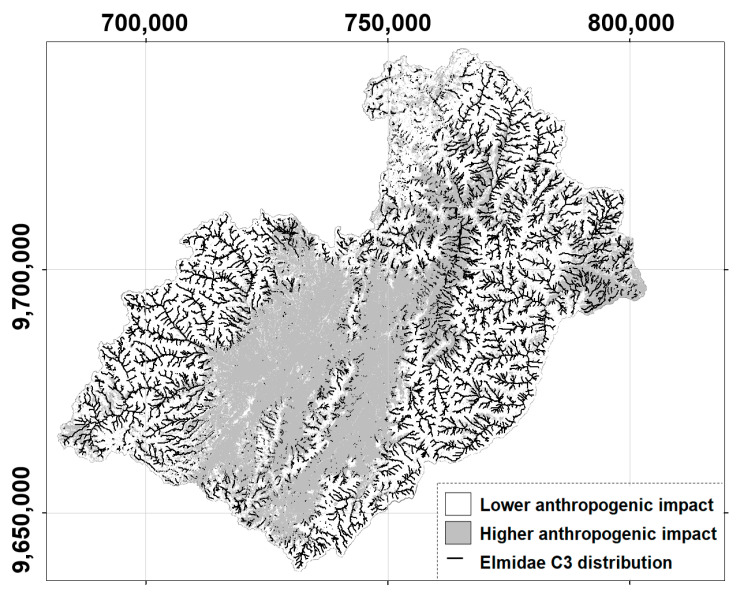
Spatial distributions of the C3 class of occurrence probability of elmid genera and estimated anthropogenic impact in the Paute river basin, Ecuador.

**Table 1 biology-12-00473-t001:** Description of the twelve environmental variables (12en_v_) used in developing the species distribution models (SDMs).

Source	Variable		Used Tool in ArcGis/ Methodology	Unit	Abbreviation	Ecological Importance	Range
	Name	Type					
DEM of SIGTIERRAS project (Corral and Montiel Olea, 2020)	Elevation	Continuous	Spatial Analyst > Hydrology > Fill	m a.s.l	Elev	Temperature tends to be colder at higher elevations, e.g., in páramo ecosystems, influencing water dissolved oxygen values [50,51,52].	411–4212
	Slope	Continuous	Spatial Analyst > Surface > Slope	Degree	Slp	Water velocity and, consequently, oxygen content, are related to slope [21].	0–74.1
	Flow direction	Categorical	Spatial Analyst > Hydrology > Flow Direction	(-)	Fdir	Flow direction is related to substrate accumulation and streambed heterogeneity [53].	1–128
	Shreve stream order	Continuous	Spatial Analyst > Hydrology > Stream Order	(-)	Shreve	High-stream order values are indicators of bigger discharges [21,54].	1–5367
	Eastness	Continuous	Spatial Analyst > Map Algebra > Raster Calculator [55]	(-)	East	These factors are related to the terrain declivity, stream course direction, and luminosity, which affect water temperature, oxygen [56], and algae growth. Algae are food sources for certain elmids [57].	−1–1
	Northness	Continuous			Ntns		
	Sinuosity	Continuous	Stream Gradient and Sinuosity > Shapefiles > Calculate Sinuosity [58]	(-)	Snty	The sinuosity is related to the accumulation of sediments and channel heterogeneity [59].	1–4.8
National Institute of Meteorology and Hydrology (http://www.inamhi.gob.ec accessed on 7 February 2022)	Precipitation	Continuous	Spatial Analyst > Map Algebra > Raster Calculator	mm	PP	Precipitation is directly related to water availability and indirectly to water velocity and oxygen content [60].	586.5–3237.7
Geopedological map, scale 1:25,000; SIGTIERRAS project (Corral and Montiel Olea, 2020)	Lithology	Categorical	Conversion > To Raster > Polygon to Raster	(-)	Ltlgy	Elements in the water and sediments of rivers are present because of the natural weathering of the surrounding lithology [61]. These elements conditionate the elmids [62].	1–78
	Soil type	Categorical	Conversion > To Raster > Polygon to Raster	(-)	Soils	Water chemistry of rivers is affected by surrounding soil units [63].	1–10
Land Use map, scale 1:100,000 (MAE, 2013)	Riparian alteration	Continuous	[64,65]	%	Rip-alt	The riparian zones regulate water temperature and allochthonous organic matter inputs and mitigate the effects of anthropogenic pressures [21,66].	0–99
Global Land Analysis and Discover (https://glad.umd.edu/dataset/ge accessed on 7 February 2022)	Canopy	Continuous	Data Management > Raster > Raster Processing > Resample	(-)	Cnpy	Canopy attenuates the sunlight, regulates the water temperature of streams and favours streambed heterogeneity [66,67].	0–100

**Table 2 biology-12-00473-t002:** Random Forest predictions for each genus of the Elmidae family. The aggregated area under curve (AUC) values are the result of considering either all the independent variables (“step 1”) or only the significant variables identified as truly important for explaining the spatial probability of occurrence of elmids (“step 2”). “*” indicates that the median was chosen as the aggregated AUC value. SDM = species distribution model. Probability of occurrence class: low (C1), medium (C2), high (C3).

	AUC		SDM of Probability of Occurrence			
Genus	Mean/Median		Probability Range	Spatial Extent (%)		
	(Step 1)	(Step 2)		C1	C2	C3
*Austrelmis*	0.76	0.83	0.00–0.94	52.8	35.4	11.9
*Austrolimnius*	0.87	0.89 *	0.00–1.00	25.0	37.4	37.7
*Heterelmis*	0.76	0.79	0.01–0.99	33.3	34.9	31.7
*Macrelmis*	0.76	0.82	0.00–0.94	28.6	41.4	30.0
*Neoelmis*	0.70	0.76	0.00–0.87	48.1	48.9	2.9

**Table 3 biology-12-00473-t003:** Environmental variables that were identified as important to explain the spatial variability of each genus of the Elmidae family. The importance value (var_imp_) for each variable is expressed in percentage (“*” indicates that the median central tendency measure was used to define the aggregated variable value). Cnpy = Canopy; Elev = Elevation; East = Eastness; Fdir = Flow direction; Ltlgy = Lithology; PP = Precipitation; Rip-alt = Riparian alteration; Shreve = Shreve stream order; Slp = Slope; Snty = Sinuosity.

Genera	Environmental Variable and Its Weight (%)					
*Austrelmis*	Elev *	PP *	East *	Slp *	Rip-alt *	
	28.92	24.56	16.90	6.45	5.57	
*Austrolimnius*	Elev	Ltlgy *	East *	Fdir *	Slp *	
	51.70	35.39	4.10	2.11	1.86	
*Heterelmis*	Elev	Ltlgy	Slp *	East	Shreve *	Rip-alt *
	52.27	26.95	6.63	5.00	3.27	2.06
*Macrelmis*	PP	Shreve	Elev *	East	Slp *	Cnpy *
	53.56	19.56	7.17	6.23	5.76	2.83
*Neoelmis*	PP	Slp	Cnpy	East *	Snty	
	47.62	10.06	9.41	6.38	5.45	

## Data Availability

Restrictions apply to the availability of data used in this research. The database belongs to the former Ecuadorian National Secretary of Water (SENAGUA), which was recently absorbed by the Ecuadorian Environmental, Water and Ecological Transition Ministry (MAATE). The SENAGUA issued the respective data use authorisation to the first author in the scope of his PhD research. This signed agreement explicitly forbids him sharing the information with a third party without their specific authorisation, which is not easy to obtain. The data might be available directly from the MAATE.

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
