# Peer review of "Occurrence Prediction of Riffle Beetles (Coleoptera: Elmidae) in a Tropical Andean Basin of Ecuador Using Species Distribution Models"

_biology, 2023, doi:10.3390/biology12030473_

Round 1

Reviewer 1 Report

In my point of view, this study used a relatively complicate method but got a a simple result of  no-general significance. It may be valuable to publish in the journal Biology, but it is better to be viewed by a strict expert on the method part. I am sorry I am not. 

Author Response

c1: In my point of view, this study used a relatively complicate method but got a simple result of  no-general significance. It may be valuable to publish in the journal Biology, but it is better to be viewed by a strict expert on the method part. I am sorry I am not.

a1: The authors would like to thank the reviewer for going through the draft of our manuscript. Nevertheless, they do not subscribe to the opinion of the reviewer in the sense that our manuscript produced only a simple non-significant result. Our research (i) used a pioneering application of a sophisticated mathematical-statistical approach; (ii) was applied on a large basin that plays a significant role for the development of Ecuador; (iii) is pioneering since there is almost no research carried out about tropical Andean basins using Species Distribution Models; (iv) used Elmidae, which is known to be sensitive to anthropogenic impact and, as such, important for water resources management; (v) shows evidence of different ecological requirements for each elmid genus echoing the complexity of the Elmidae family.

Reviewer 2 Report

This is a very interesting paper on predicting the occurrence of Elmidae (Riffle Beetles) in tropical aquatic habitats of Ecuador. The authors implemented Species Distribution Models with specific adjustments like  Random Forest (RF) algorithm to analyze a large dataset, collected in the field. As a result, the authors identify several environmental variables that explained most of the Elmidae spatial variability, as well indicate species-specific habitat selectivity, expressed by the different probability of occurrence. These results could be implemented in conservation actions in the study area.

 In my opinion, the manuscript is well written and deserves publication in „biology” without major revision. Below I raise several questions to hear the opinion of authors:

 How to understand the sentence: „Herein, to use a record of an elmid genus, such to perform the modeling process, the minimal sample size (Cao et al., 2013) was greater than two individuals” (lines 187-189) – is it two individuals of a given elmid genus, or total sample of two individuals? If „two individuals of a given elmid genus”, why not one individual?

 Did authors consider the issue of „false absences” (eg not recording the genus due to stochastic events) in the field data? Does the analysis the of probability occurrence compensate for the problems of detecting rare organisms? – I am very interested to see the justification and literature on this issue.

 The authors had chosen Species Distribution Models to analyze the presence/absence data in regard to environmental variables. Does this approach have an advantage over General Linear Models widely used the in the evaluation of habitat selection of organisms?

 The authors describe in detail 12 environmental variables (chapter 2.3. Environmental variables, including theTable 1), however, in the later analysis (chapter 3.2. C3 class of occurrence probability of Elmidae across the Paute river basin) they introduce an additional factor „anthropic impact”. Please provide: how was this factor estimated? what are the components of „anthropic impact”? Please also consider replacing „anthropic” with „anthropogenic”. The term „anthropic” has an additional meaning related to the effect of the observer on the ongoing experiment, and „anthropogenic” clearly indicates the effect of human activities on the environment.

Author Response

c1: This is a very interesting paper on predicting the occurrence of Elmidae (Riffle Beetles) in tropical aquatic habitats of Ecuador. The authors implemented Species Distribution Models with specific adjustments like Random Forest (RF) algorithm to analyze a large dataset, collected in the field. As a result, the authors identify several environmental variables that explained most of the Elmidae spatial variability, as well indicate species-specific habitat selectivity, expressed by the different probability of occurrence. These results could be implemented in conservation actions in the study area.

In my opinion, the manuscript is well written and deserves publication in „biology” without major revision.

a1: The authors would like to thank the reviewer for his/her effort to enhance the quality of our manuscript. We also appreciate very much his/her appreciation of the report of our research.

  c2: How to understand the sentence: „Herein, to use a record of an elmid genus, such to perform the modelling process, the minimal sample size (Cao et al., 2013) was greater than two individuals” (lines 187-189) – is it two individuals of a given elmid genus, or total sample of two individuals? If „two individuals of a given elmid genus”, why not one individual?

a2: In our study, for every one of the sampling stations, we considered at least two individuals per taxa (recorded in each station). We decided to proceed in this way to minimise the probability that an individual of a given taxa was recorded accidentally (i.e., accidental arrival through strong current; or a dead individual drifted downwards by the river current; etc.) in the sampling station of interest. This explanation was included in the newer version of our manuscript, in section 2.2.1.

  c3: Did authors consider the issue of „false absences” (eg not recording the genus due to stochastic events) in the field data? Does the analysis of probability occurrence compensate for the problems of detecting rare organisms? – I am very interested to see the justification and literature on this issue.

a3: The following answer is intended only for the reviewer. It has not been incorporated in the revised version of our manuscript.

 (Numerical) modelling predictions are always dependant on the quality of the data with which models are being fed as well as on the degree of complexity of the modelling tool; there should be a trade-off among these aspects (Grayson and Blöschl, 2000). Uncertainty is always present in any model application. It might arise from different sources that could be grouped into the following three classes (Gupta et al., 1998; Vázquez et al., 2009; Vázquez and Hampel, 2022): (i) data; (ii) model structure; and (iii) scale of application. These different sources of uncertainty are difficult to address explicitly and accurately; indeed, there is a lot of discussion going out there on the best way of accounting for uncertainty in modelling and reducing it (Gupta et al., 1998; Beven et al., 2007; Elith and Leathwick, 2009; Vázquez and Hampel, 2022). Assuming that our data is perfect (and that no major scaling issues are present), modelling uncertainty is normally reduced to imperfections implicit in the modelling tool. This is the implicit assumption that is lying beneath most of modelling activities. However, even under this assumption, despite adoption of very robust mathematical and statistical approaches (and accurate descriptions of modelled processes), these model imperfections exist, because models are just (imperfect) representations of reality.

 Thus, cancelling modelling uncertainty is virtually impossible; at least not with the means and knowledge that we possess right now. Rather, what should be pursued is to minimise as much as feasible the above-mentioned sources of uncertainty. Hereafter, adopting monitoring strategies that reduce data uncertainty, using advanced and powerful mathematical and statistical modelling tools, and applying them congruently with the scale of the study, is what should be done in every (particularly, research) modelling application. This is what we have tried to accomplish in our study. On one hand, we have tried to minimise the uncertainty implicit in our data by embracing some strategies, such as, ignoring data that are not above a minimum number of individuals or revisiting as much as feasible the monitoring sites (i.e., carrying out as many replicates as feasible).

 Nevertheless, despite our efforts for minimising this uncertainty, we are aware that it is still present in our data (as in the data of any other prior or future modelling study to be carried out by other research teams, elsewhere). Simply, there are many economical and practical constraints, including deficiencies of standard sampling methods, that impede achieving a more ideal data uncertainty reduction; and this is even worse in remote and topographically-difficult-to-monitor study sites, such as the one of our study. Furthermore, accounting accurately for this data uncertainty is rather difficult to achieve in any study, particularly when working with benthic macroinvertebrates. Taking samples under these conditions generally reflects/represents only a tiny fraction of the overall conditions in the sampling site (and water body). Thus, any inferences drawn from the macroinvertebrates data are, to a certain extent, uncertain (McBride, 2005).

 Hence, the probability of not recording some Elmid genera that are normally present in a site, owing to some potential stochastic events, exists, and, most likely, it is also connected to the previously referred deficiencies of standard sampling methods. Definitively, estimating this particular type of data uncertainty would not be trivial. A specific and well-planned monitoring strategy (i.e., multi-site, multi-sampling method, implying different surveyors, transient...) should be applied to assess, at least approximately, the magnitude of the probability of false absences. And even then, that probability estimate will be not only incomplete but also study-site-specific. This was definitively beyond the scope of our manuscript. This is the subject of an uncertainty analysis (Fayyad et al., 1996; Diekmann and Featherman, 1998; Guisan and Zimmermann, 2010; Elith and Leathwick, 2009; Beven, 2010; Sinclair et al., 2010; Vázquez and Hampel, 2022), that is, of a specific research by itself. For the same reason, we did not attempt either a theoretical study (i.e., even less accurate than an experimentally based assessment) upon some data probability scenarios.

We believe that the above mentioned measures adopted from our side, together with the performed high temporal and spatial sampling efforts, are facts that contributed importantly to reduce data uncertainty; that is, despite the potential stochastic affection of monitoring, a large record of sampled Elmidae is likely to have reduced this probability to acceptable levels. Further, we are convinced that the adoption of the reported sophisticated mathematical and statistical approach has contributed to minimise the uncertainty attached to our modelling tool. Nevertheless, it must be stated that the analysis of predicted occurrence probability does not completely compensate the problems linked with rare organisms. This problem was most likely (partially) compensated throughout the training phase of this modelling; that is, it is likely that the model parameter values would be, to a certain extent, different if a perfect data set was available.

 CITED REFERENCES

Beven, K., Smith, P., Freer, J., 2007. Comment on ‘‘Hydrological forecasting uncertainty assessment: Incoherence of the GLUE methodology’’ by Pietro Mantovan and Ezio Todini, Journal of Hydrology, pp. 315-318.

Beven, K. Environmental modelling: an uncertain future?. CRC press, 2010.

Elith, J., Leathwick, J.R., 2009. Species Distribution Models: Ecological Explanation and Prediction Across Space and Time. Annual Review of Ecology, Evolution, and Systematics 40, 677-697.

Diekmann, J. E., & Featherman, W. D. (1998). Assessing cost uncertainty: Lessons from environmental restoration projects. Journal of construction engineering and management, 124(6), 445-451.

Fayyad, U., Piatetsky-Shapiro, G., & Smyth, P. (1996). From data mining to knowledge discovery in databases. AI magazine, 17(3), 37-37.

Guisan, A., & Zimmermann, N. E. (2000). Predictive habitat distribution models in ecology. Ecological modelling, 135(2-3), 147-186.

Grayson, R., Blöschl, G., 2000. Spatial Patterns in Catchment Hydrology Observations and Modelling. Cambridge University Press, Cambridge.

Gupta, H.V., Sorooshian, S., Yapo, P.O., 1998. Toward improved calibration of hydrologic models: multiple and non-commensurable measures of information. Water Resources Research 34, 751-763.

McBride, G. B. (2005). Using Statistical Methods for Water Quality Management - Issues, Problems and Solutions. Retrieved from https://www.ptonline.com/articles/how-to-get-better-mfi-results

Sinclair, Steve J., Matthew D. White, and Graeme R. Newell. "How useful are species distribution models for managing biodiversity under future climates?." Ecology and Society 15.1 (2010).

Vázquez, R.F., Beven, K., Feyen, J., 2009. GLUE based assessment on the overall predictions of a MIKE SHE application. Water Resources Management 23, 1325-1349.

Vázquez, R.F., Hampel, H., 2022. A Simple Approach to Account for Stage-Discharge Uncertainty in Hydrological Modelling. Water 14, 1045.

c4: The authors had chosen Species Distribution Models to analyze the presence/absence data in regard to environmental variables. Does this approach have an advantage over General Linear Models widely used in the evaluation of habitat selection of organisms?

 a4: Species distribution models (SDMs) are numerical tools that combine observations

of species occurrence or abundance with environmental estimates. Indeed, General Linear Models (GLMs) are widely used to investigate the individual or combined effects of environmental factors on the suitable habitat of organisms; that is, for developing SDMs (Elith and Leathwick, 2009). This regression technique can explain and predict the presence or abundance of species as a function of one or many site factors (elevation, slope, aspect, solar radiation, precipitation, soil type, etc.).

 Nevertheless, the input datasets of the species distribution models (SDMs) are imbalanced, which is a problem for methods that cannot deal with a lack of proportionality between presence and absence data (i.e., imbalanced data set) (Johnson et al., 2012).  Further, another typical problem in the development of SDMs has to do with the lack of enough data (Mi et al., 2017), which, in general, is a redundant problem in ecology. RF is a method that can deal with both problems (Khalilia et al., 2011; Brown et al., 2012; Larras et al., 2017; Mi et al., 2017). Further, RF has few prior assumptions about error distributions; and, in many cases, it outperforms the GLM method in SDM applications (Williams et al., 2009; Li & Wan, 2013; Rodriguez-Galiano et al., 2015). These are the reasons why we decided to use RF and not GLM or GAM or CART methods for our study.

 CITED REFERENCES

Brown, I. & Mues, C. An experimental comparison of classification algorithms for imbalanced credit scoring data sets. Expert Syst. Appl. 39, 3446–3453 (2012).

 Elith, J., Leathwick, J.R., 2009. Species Distribution Models: Ecological Explanation and Prediction Across Space and Time. Annual Review of Ecology, Evolution, and Systematics 40, 677-697.

Johnson, R. A., Chawla, N. V. & Hellmann, J. J. Species distribution modelling and prediction: A class imbalance problem. Proc. - 2012 Conf. Intell. Data Understanding, CIDU 2012 9–16 (2012). doi:10.1109/CIDU.2012.6382186

Khalilia, M., Chakraborty, S. & Popescu, M. Predicting disease risks from highly imbalanced data using random forest. BMC Med. Inform. Decis. Mak. 11, (2011).

Larras, F. et al. Assessing anthropogenic pressures on streams: A random forest approach based on benthic dia-tom communities. Sci. Total Environ. 586, 1101–1112 (2017).

Li, Xinhai, and Yuan Wang. Applying various algorithms for species distribution modelling. Integrative zoology 8.2 (2013): 124-135.

Mi, C., Huettmann, F., Guo, Y., Han, X. & Wen, L. Why choose Random Forest to predict rare species distribution with few samples in large undersampled areas? Three Asian crane species models provide supporting evidence. PeerJ 2017, (2017).

Rodriguez-Galiano, V., Sanchez-Castillo, M., Chica-Olmo, M., & Chica-Rivas, M. J. O. G. R. (2015). Machine learning predictive models for mineral prospectivity: An evaluation of neural networks, random forest, regression trees and support vector machines. Ore Geology Reviews, 71, 804-818.

Williams, J. N., Seo, C., Thorne, J., Nelson, J. K., Erwin, S., O’Brien, J. M., & Schwartz, M. W. (2009). Using species distribution models to predict new occurrences for rare plants. Diversity and Distributions, 15(4), 565-576.

 c5: The authors describe in detail 12 environmental variables (chapter 2.3. Environmental variables, including the Table 1), however, in the later analysis (chapter 3.2. C3 class of occurrence probability of Elmidae across the Paute river basin) they introduce an additional factor „anthropic impact”. Please provide: how was this factor estimated? what are the components of „anthropic impact”? Please also consider replacing „anthropic” with „anthropogenic”. The term „anthropic” has an additional meaning related to the effect of the observer on the ongoing experiment, and „anthropogenic” clearly indicates the effect of human activities on the environment.

a5: Once more, the authors would like to take this opportunity for thanking to the reviewer for his/her very constructive comments and suggestions. The authors agree with the reviewer in that there is an important difference between the terms ”anthropic“ and “anthropogenic“. Thus, following his advice, the authors have adopted the term ”anthropogenic” throughout the newer version of the manuscript.

 The explanation on the determination of the spatial extent of classes lower/higher anthropogenic impact zones was given in the original version of the manuscript in section 2.6. “Congruency of the predicted spatial distribution of the C3 probability of occurrence of elmid genera”. After the comment of the Reviewer, this explanation has been enhanced by adding more details on the GIS operations that were performed for deriving these classes. The additional details are as follows. Basically, the original LU-LC classes (MAE, 2013) were the following: (1) altered vegetation; (2) woody native vegetation; (3) without cover/urbanised, (4) páramo ecosystem; and (5) water. The higher anthropogenic impact class was defined upon the reclassification of LU-LC classes 1 and 3, whilst the lower anthropogenic impact class was defined upon classes 2, 4, 5. Thus, these anthropogenic impact classes are not the result of any additional calculation of an index or a factor but just a simple reclassification of the original LU-LC information.

 CITED REFERENCES

MAE. Sistema de clasificación de ecosistemas del Ecuador continental. (Ministerio del Ambiente del Ecuador, Subsecretaría de Patrimonio Natural - Proyecto mapa de vegetación, 2013).

Reviewer 3 Report

This is a nice paper showing the results of a rather underused method for creating species distribution models, one that can incorporate not just presence records, but absence data. Where available and robust, such data and methods that take absence into account offer an upgrade over more widely used MaxEnt methods.

Honestly, much of the modelling discussion was a bit over my head, although it all seemed reasonable and well-argued. My one major concern with this study, though not the RF approach in general, is this modelling based on genera rather than species. I understand that for larvae, it is difficult or impossible to go further, but that is a significant limitation. Various water quality monitoring papers over the years have stressed the importance of species identification, as congeneric species in many, perhaps most, genera can vary widely in their preferences and tolerances. Some species in a genus may correspond to the models inferred here, but not all. And creating a model based on a random assortment of congeners will have less power and precision than one based on particular species. This is weakly alluded to in the previous section, but probably should receive greater attention throughout, since the authors promote their approach pretty strongly.

Otherwise, I made only a few minor corrections and comments in the text that can be easily addressed.

Author Response

c1: This is a nice paper showing the results of a rather underused method for creating species distribution models, one that can incorporate not just presence records, but absence data. Where available and robust, such data and methods that take absence into account offer an upgrade over more widely used MaxEnt methods.

Honestly, much of the modelling discussion was a bit over my head, although it all seemed reasonable and well-argued.

a1: The authors would like to thank the reviewer for going through the draft of our manuscript and for his/her very positive and constructive comments and suggestions.

  c2: My one major concern with this study, though not the RF approach in general, is this modelling based on genera rather than species. I understand that for larvae, it is difficult or impossible to go further, but that is a significant limitation. Various water quality monitoring papers over the years have stressed the importance of species identification, as congeneric species in many, perhaps most, genera can vary widely in their preferences and tolerances. Some species in a genus may correspond to the models inferred here, but not all. And creating a model based on a random assortment of congeners will have less power and precision than one based on particular species. This is weakly alluded to in the previous section, but probably should receive greater attention throughout, since the authors promote their approach pretty strongly.

a2: Once more the authors would like to thank the reviewer for his/her efforts to enhance the quality of our manuscript.

 The authors agree with the Reviewer that the best would be identifying the different organisms to the species level. As the Reviewer rightly mentions, in most of the neotropical regions, there is not enough knowledge to reach the species taxonomic level in the case of river macroinvertebrates. We consider that in such a case, it is preferable to have well identified individuals at genus level than having potential high error rates of classification at species level. The Reviewer is right as well in the sense that congeners within the genus could have different environmental preferences, which could reduce modelling power. Hence, we added a text in the discussion section (4.2) of the newer version of the manuscript, describing this concern.

 However, Elmidae family is a good bioindicator and all the study genera (and, most likely, the respective species) prefer clean and well-conserved streams (Elliott, 2008; Garcia-Criado and Fernandez-Alae, 2001; Von Ellenrieder, 2007) despite the different ecological requirements identified in the present study. Hence, we consider that identification on genus level is sufficient to contribute to local conservation and restoration efforts. Using genera of Elmidae in SDM we could identify areas heavily impacted by anthropogenic activities and areas where restoration projects could be executed in the future. Hereafter, we believe that despite using the genus level, our methodology could be used in similar catchments for conservation purposes where water resource managers and technicians face the same difficulties of taxonomic identification as we did.

 CITED REFERENCES

Elliott, J. M. The Ecology of Riffle Beetles (Coleoptera: Elmidae). Freshw. Rev. 1, 189–203 (2008).

Garcia-Criado, F. & Fernandez-Alaez, M. Hydraenidae and Elmidae assemblages (Coleoptera) from a Spanish river basin: good indicators of coal mining pollution? Arch. Fur Hydrobiol. 150, 641–660 (2001). 

Von Ellenrieder, N. Composition and structure of aquatic insect assemblages of Yungas mountain cloud forest streams in NW Argentina. Rev. la Soc. Entomológica Argentina 66, 57–76 (2007). 

c3: In lines 17 and 21 replace picks by peaks.

a3: The amendments were implemented in the newer version of the manuscript.

c4: In lines 179 and 180, this is awkward - I'd rephrase as "the possibility that different congeners differ in habitat preferences could explain..."

a4: The reviewers agree with the suggestion of the reviewer. The original text was rephrased. Once more, thanks very much for your efforts to enhance the contents and shape of our manuscript.